

# Rapidity distribution within the defocusing non-linear Schrödinger equation model

Yasser Bezzaz, Léa Dubois and Isabelle Bouchoule⋆

Laboratoire Charles Fabry, Institut d'Optique Graduate School, CNRS,
Université Paris-Saclay, 91127 Palaiseau, France

⋆ isabelle.bouchoule@institutoptique.fr

## Abstract

We consider the classical field integrable system whose evolution equation is the non-linear Schrödinger equation with defocusing non-linearities, which is the classical limit of the quantum Lieb-Liniger model. We propose a simple derivation of the relation between two sets of conserved quantities: on the one hand the trace of the monodromy matrix, parameterized by the spectral parameter and introduced in the inverse-scattering framework, and on the other hand the rapidity distribution, a concept imported from the Lieb-Liniger model. To do so we use the definition of the rapidity distribution as the asymptotic momentum distribution after a very large expansion. We propose two different ways to derive the result, each one using a thought experiment that implements an expansion.

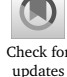

# 1 Introduction

The Lieb-Liniger model, that describes one-dimensional Bosons with contact repulsive inter-actions [1], plays a key role in quantum many body systems. On the experimental point of view, it describes remarkably well cold-atoms experiments (see for instance the review [2]), among them the famous Newton's Craddle experiment [3]. On the theoretical point of view, it is a paradigmatic integrable model, that is the non-relativistic limit of all known integrable quantum field theories [4, 5]. The integrability manifests itself by the fact that the eigenstates take the form of Bethe-Ansatz wave functions. The latter are labeled by numbers, whose unit is mass×velocity, and whose number is equal to the particles number, called the rapidities or the Bethe-roots. For a large system, one defines the coarse-grained rapididty distribution $\Pi(p)$ as the density of rapidities: $\Pi(p)\mathrm{d}p$ is the number of Bethe-roots in the interval $[p, p + \mathrm{d}p]$. By construction, it is a conserved quantity. Moreover, for a system confined on a length $L$, its intensive counterpart $\rho(p) = \Pi(p)/L$ plays a crucial role in the long time behavior: as long as mean values of local quantities are concerned, the system shows a relaxation phenomena and the relaxed system is entirely characterized by $\rho(p)$ [2, 6, 7]. Many results have been obtained in recent years for relaxed states, expressing mean values of local operators in terms of $\rho(p)$ [8, 9].[1] The fact that relaxed states are entirely parameterized by $\rho(p)$ is also at the heart of the Generalized Hydrodynamics theory, that assumes local relaxation [2, 12, 13].

A very famous asymptotic description of the Lieb-Liniger model is the classical field descrip-tion which ignores quantization of the particles and describes the system as a classical field $\psi(x)$, where $\psi$ is a complex field and $x$ is the spatial coordinate [2, 14–16]. The time evolution of $\psi(x)$ is given by the nonlinear Shrödinger equation (NLSE), also called the Gross-Pitaevskii equation. The classical field description has proven to be extremely powerful in describing many experimental results in the field of cold atoms experiments [15]. It also successfully de-scribes many other experiments such as propagation of light in a non-linear medium [17]. The NLSE belongs to the class of classical integrable models which have been the subject of a whole domain of mathematical physics since the 1960's. The inverse scattering method enables to construct an infinite set of independent conserved quantities, parameterized by a spectral pa-rameter $\lambda$, called inverse scattering constants of motion in the rest of this paper and denoted $\tau_\lambda$ [18]. Importantly, these constants of motion can be computed at any time, provided that the field configuration at this particular moment is known.

Making the connection between the classical and the quantum framework is a highly de-sirable task as it enables to extend recent results obtained for relaxed states of the Lieb-Liniger model to the classical framework. One needs for this to identify the classical counterpart of the rapidity distribution and to express it in terms of the inverse scattering constants of mo-tion. To do so one can use the very powerful Quantum Inverse Scattering Method (QISM), the link with the Bethe-Ansatz rapidities being done via the Algebraic Bethe-Ansatz method [19]. This task has been done for the sinh-Gordon model in [20] and more recently in [21] for the Lieb-Liniger model. These results made it possible to generalize calculations of correlation functions in relaxed states of the quantum model to the classical framework [20, 22], and to identify the classical counterpart of the Generalized Hydrodynamics theory [21].

The results cited above use very advanced mathematical techniques. In this paper, we propose on the contrary a very simple way to extend the notion of rapidity distribution to the classical framework and we propose a simple and elementary derivation of the link between the rapidity distribution and the inverse scattering constants of motion. For this, we will not rely on the definition of the rapidity distribution based on the Bethe-Ansatz form of the eigen-states of the Lieb-Liniger model. Instead, we use the fact that the rapidity distribution is the asymptotic momentum distribution of the Bosons after their expansion to very large distances,

---

[1]Note also related work in another quantum integrable model in [10] and [11].

a property which provides an alternative definition of the rapidity distribution [2]. This definition of the rapidity distribution can also apply within the classical field framework: the notion of expansion is of course meaningful within the classical model, and the momentum distribution of the Bosons is nothing else, in the classical field framework, but the field density in Fourier space. The fact that, upon expansion on sufficiently large distances, the momentum distribution reaches a stationary asymptotic function is not a surprise: once diluted enough, the non-linear terms, which are the classical field counterpart of the interactions in the many-body quantum model, become negligible and the momentum distribution no longer evolves in time. What is very special about integrability is that this asymptotic momentum distribution, which is the rapidity distribution, does not depend on the time at which the expansion is performed, even though a complex dynamic could occur in the system prior to the expansion.

Using thought experiments that exploit the above definition of the rapidity distribution, we derive the link between the rapidity distribution and the inverse scattering constants of motion: more precisely, we express the inverse scattering constants of motion in terms of the rapidity distribution. For pedagogical purposes, we propose two different derivations in this paper, both related to different thought experiments and leading to different mathematical approaches.

## 2  Main result

We consider the classical field description of 1D Bosons of mass $m$ with contact repulsive interactions. The system is described by the one-dimensional complex field $\psi(x)$, that fulfills the Poisson-Bracket relations $\{\psi(x), \psi^*(x')\} = i\delta(x - x')/\hbar$, $\{\psi(x), \psi(x')\} = 0$ and whose Hamiltonian is

$$H = \frac{\hbar^2}{2m} \int_0^L \mathrm{d}x \left| \frac{\partial \psi}{\partial x} \right|^2 + \frac{g}{2} \int \mathrm{d}x \, |\psi(x)|^4 \,, \tag{1}$$

where $g$, which governs the non-linear term, is the coupling constant. Here we assume periodic boundary conditions on the box of length $L$. The equation of motion of $\psi$ is the NLSE

$$i\hbar \frac{\partial \psi}{\partial t} = -\frac{\hbar^2}{2m} \frac{\partial^2 \psi}{\partial x^2} + g|\psi|^2 \psi \,. \tag{2}$$

In the following, to lighten the notations, we use a unit system in which $\hbar = m = 1$. The Fourier components of $\psi$ are $\psi_k = \int_0^L \mathrm{d}x \psi(x)e^{-ikx}/\sqrt{L}$ where $k$ takes the discrete values which are the multiples of $2\pi/L$ and one defines the momentum distribution as the continuous function

$$n(p) = \frac{L}{2\pi} \langle |\psi_k|^2 \rangle_{\text{c.g.}} \,, \tag{3}$$

where the right-hand-side is computed for $k$ values close to $p$ and c.g. means coarse-graining on a width in $k$ small compared to the width in $p$ of $n(p)$ but sufficient to wash out fluctuations of $\psi_k$ that may occur on a small scale in $k$ space. It is normalized by $\int \mathrm{d}p \, n(p) = \int \mathrm{d}x |\psi(x)|^2$. Note that the weights $|\psi_k|^2$ are not constants of motion since interactions mix different Fourier components, and the function $n(p)$ evolves in time in general.

The integrability of the NLSE is manifested by the fact that the *asymptotic* momentum distribution after a very long expansion, $n_\infty(p)$, is a conserved distribution, in the sense that it does not depend on the time at which the expansion is performed. As explained in the introduction, this conserved distribution is nothing else but the rapidity distribution, $\Pi(p)$, namely

$$\Pi(p) = n_\infty(p) \,. \tag{4}$$

This equality provides a definition of the rapidity distribution, which is that used in this paper. The values $\Pi(p)$, labeled by the momentum $p$, define an infinite set of constants of motion.

The inverse scattering method provides an alternative set of constants of motion [19], denoted $\tau_\lambda$, labeled by a real parameter $\lambda$ called the spectral parameter, whose unit is a momentum. More precisely, $\tau_\lambda$ is the trace of the monodromy matrix, itself parametrized by $\lambda$, whose definition is recalled in section 4. The constants $\tau_\lambda$ can be computed at any time, knowing the field configuration $\psi(x)$ at this time. In the following, for the calculation of the inverse scattering constants of motion, we consider a quantization box of length $L$ large enough so that the momentum distribution of the gas, if it expanded in this box, would have converged towards its rapidity distribution. The gas being initially confined in a smaller box of size $L_0$, one extends the initial field configuration to the box of size $L$ by setting $\psi(x) = 0$ outside the box of size $L_0$. The goal of this paper is to establish the link between the inverse scattering constants of motion and the rapidity distribution. Our result is

$$\tau_\lambda = 2e^{\pi\Pi(\lambda)gm/\hbar}\cos\left(\frac{\lambda L}{2\hbar} + \frac{mg}{\hbar}\fint\frac{\Pi(p)\mathrm{d}p}{p-\lambda}\right),\tag{5}$$

where $\fint$ means the Cauchy principal value and we reintroduced $\hbar$ and $m$ for more clarity. This expression is compatible with the results obtained in [21] by taking the semi-classical limit of formulas derived from the QISM and the Algebraic Bethe-Ansatz, provided that we go to the thermodynamic limit. As expected, for large $\lambda$ the famous trace identities are recovered [19].[2] A similar expression was derived for the Sh-Gordon model in [20] (see Eq. (421) and (424) of [20]) using classical limit of Bethe-Ansatz equations. Eq. (5) also coincides with the formula (76) of [22] at large $\lambda$.

Eq. (5) shows that, for a given rapidity distribution $\Pi(p)$ and a large box length $L$, the inverse scattering constants of motion oscillate rapidly with $\lambda$. Such oscillations are smeared out if one considers the coarse-grained quantity $\langle\tau_\lambda^2\rangle_{\mathrm{c.g.}}$, where coarse-graining is done on a width large compared to $1/L$. Eq. (5) then leads to

$$\langle\tau_\lambda^2\rangle_{\mathrm{c.g.}} = e^{2\pi\Pi(\lambda)gm/\hbar},$$

a quantity which no longer depends on the size of the quantization box.

## 3 Sketch of the derivation

As advertised in the introduction, we propose two different methods to derive Eq. (5). They are based on two different thought experiments, depicted in Fig.1. The first method assumes relaxation of the system in a large box while in the second method, we consider an expansion of the system to the far-field regime. In both thought experiments, in its final state, the gas has expanded sufficiently so that its momentum distribution has converged towards its rapidity distribution.

The inverse scattering constants of motion $\tau_\lambda$ are computed from the knowledge of the field configuration $\psi(x)$, at a given time. Since they are preserved by the dynamics, one can choose to estimate them after the expansion, which is what we do in this paper. For each thought experiment, we use a dedicated technique to express the constants of motion $\tau_\lambda$ in terms of the momentum distribution of the field. Since the latter is nothing else but the rapidity distribution, we thus obtain an expression relating the inverse scattering constants of motion $\tau_\lambda$ to the rapidity distribution. As it should, the calculations for both thought experiments lead to the same result, which is the one given in Eq. (5).

---

[2]This is shown taking the limit $\lambda \to i\infty$, using $\Pi(\lambda) \simeq 0$ and expanding $1/(\lambda-p)$ in power of $p/\lambda$ to evaluate the integral in the cosinus.

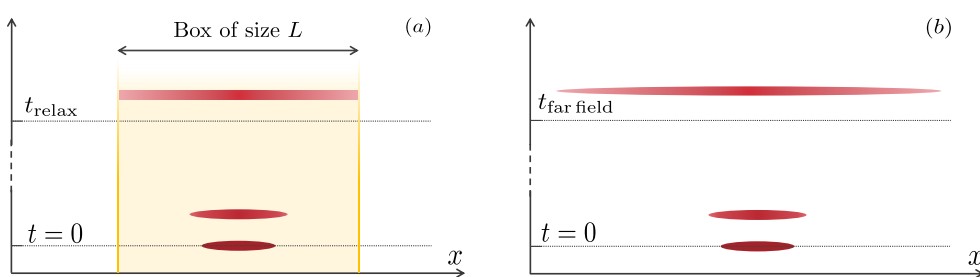

Figure 1: Naïve illustration of the thought experiments used in this paper to relate the rapidity distribution to the inverse scattering constants of motion $\tau_\lambda$, *i.e.* to derive Eq. (5). In both thought experiments, the field undergoes an expansion that we assume large enough so that the momentum distribution of the system after the expansion has converged towards the rapidity distribution. The red regions schematically represent $|\psi(x)|^2$ at three different times: just before the expansion, at the beginning of the expansion and after the expansion. The constants $\tau_\lambda$ are computed from the monodromy matrix evaluated for the field after the expansion. $(a)$: at $t = 0$, we let the system expand and relax to a very large box of size $L$. The key point of the calculation is the use of a Markovian approximation, valid since the field amplitude is very small (see section 5). $(b)$: at $t = 0$, we let the system expand freely. We consider expansions large enough to reach the far field regime in which not only the momentum distribution has converged towards the rapidity distribution but the density distribution has become homothetic to the rapidity distribution. We then compute the inverse scattering constants of motion using a calculation similar to the one made to derive the Landau-Zener formula (see section 6).

In the following sections, details of the calculation are shown. We first recall how the inverse scattering constants of motion $\tau_\lambda$ are constructed. We then present the heart and the most technical part of our derivations, namely the calculation of $\tau_\lambda$ for a a system that has expanded on a sufficiently large zone. The first derivation, based on the thought experiment shown in Fig. 1(a), uses a Markovian approximation to compute $\tau_\lambda$. The second derivation, based on the thought experiment shown in Fig. 1(b), uses a calculation similar to what is done to extract the Landau-Zener formula.

## 4 The inverse scattering constants of motion

We consider a field $\psi(x, t)$ whose time evolution is given by the NLSE Eq. (2) and which obeys periodic boundary conditions on a box of length $L$. Within the framework of the inverse scattering method, it is possible to construct an infinite set of constants of motion labeled by a spectral parameter $\lambda$. At any time $t$, one can compute these constants of motion knowing the field configuration at the time $t$. Thus in the following we consider the one-dimensional function $x \to \psi(x, t)$ and we omit the time variable. We first introduce the $2 \times 2$ matrix $T_\lambda(x)$, called the propagator, which fulfills $T_\lambda(0) = \mathrm{Id}$ and which obeys the evolution equation

$$\partial_x T_\lambda(x) = V_\lambda(x) T_\lambda(x), \tag{6}$$

where the matrix $V_\lambda(x)$ depends on $\psi(x)$ according to

$$V_\lambda(x) = \begin{pmatrix} -i\frac{\lambda}{2} & \sqrt{g}\,\psi^*(x) \\ \sqrt{g}\,\psi(x) & i\frac{\lambda}{2} \end{pmatrix}. \tag{7}$$

The propagator computed on the total length of the box, $T_\lambda(L)$, is called the monodromy matrix. The monodromy matrix depends on time via the time dependence of $\psi(x)$. However, for a field $\psi$ that obeys the NLSE (2) with periodic boundary conditions, the monodromy matrix has the remarkable property that its trace is time-independent, namely

$$\tau_\lambda = \text{Tr}(T_\lambda(L)) \tag{8}$$

is time independent [19]. The quantities $\tau_\lambda$ thus constitute a set of constants of motion, labeled by the spectral parameter $\lambda$ and denoted inverse scattering constants of motion in this paper. Note that since, upon exchange of rows and columns, $V_\lambda$ becomes its complex conjugate, the diagonal entries of $T_\lambda(x)$ are complex conjugate, the same being true for the off-diagonal entries.

Eq. (6) recalls the one obeyed by the evolution operator in quantum physics, where $x$ plays the role of time and $iV_\lambda(x)$, although it is not hermitian, plays the role of the time-dependant Hamiltonian. Inspired by this similarity, we will use, for the estimation of the monodromy matrix, techniques similar to those developed in quantum physics.

## 5 Calculation assuming relaxation in a very large box

In this section we consider the thought experiment depicted in Fig. 1(a), namely we assume the gas has expanded and relaxed to a very large box of length $L$, large enough so that the momentum distribution is equal to the rapidity distribution. To compute the inverse scattering constants of motion we will calculate the monodromy matrix using the properties of the field after relaxation in the box of size $L$.

The relaxed system is time-independent and spatially homogeneous in the following sense: if $f(\{u_i\}, x, t)$ is an N-points correlation function of the field at positions $x, x+u_1, \dots, x+u_{N-1}$, the time-averaged quantity $\langle f(\{u_i\}, x, t)\rangle = \lim_{\tau\to\infty} \int_0^\tau dt' f(\{u_i\}, x, t+t')/\tau$, where the asymptotic value is reached as soon as $\tau$ is much larger than the correlation time of the field, is independent of $x$ and $t$. In particular, $\langle \psi^*(x+u, t)\psi(x, t)\rangle$ is independent of $x$ and $t$. Moreover, the time-average of $\psi(x, t)$ vanishes.

The crucial point for the calculation of $\tau_\lambda$ is that, since it is time independent, it can be computed via Eq. (8) using the monodromy matrix at any time $t$. It implies in particular that $\tau_\lambda = \text{Tr}(\langle T_\lambda(L)\rangle)$ where averaging of the monodromy matrix is done over time. This is why in the following we compute the averaged propagator $\langle T_\lambda(x)\rangle$.[3]

Let us first go to the interaction picture by considering $\tilde{T}_\lambda = T_{0,\lambda}^{-1} T_\lambda$, where $T_{0,\lambda} = e^{-i\lambda x\sigma_z/2}$ is the propagator in the case of a vanishing field. Here $\sigma_z$ is the Pauli matrix. Then, the evolution equation (6) becomes $\partial_x \tilde{T}_\lambda(x) = \tilde{V}_\lambda(x)\tilde{T}_\lambda(x)$ with

$$\tilde{V}_\lambda(x) = \begin{pmatrix} 0 & \sqrt{g}e^{i\lambda x}\psi^*(x) \\ \sqrt{g}e^{-i\lambda x}\psi(x) & 0 \end{pmatrix}. \tag{9}$$

Let us consider the modification of the propagator from a position $x$ to a position $x + dx$. The evolution equation gives

$$\tilde{T}_\lambda(x+dx) = \tilde{T}_\lambda(x) + \int_x^{x+dx} dx' \tilde{V}_\lambda(x')\tilde{T}_\lambda(x) + \int_x^{x+dx} dx' \tilde{V}_\lambda(x') \int_x^{x'} dx'' \tilde{V}_\lambda(x'')\tilde{T}_\lambda(x''). \tag{10}$$

---

[3]Note that, since $T_\lambda(x)$ is a functional of the field $\{\psi(y)\}$, and depends on time only via the time-dependence of $\{\psi(y)\}$, the time-averaged propagator $\langle T_\lambda(x)\rangle$ is also equal to the propagator averaged over the field configurations $\{\psi(y)\}$, the weight of a configuration being equal to the proportion of time the system spends in this particular configuration during its time evolution. Thus, for the following calculations, one is free to think of averaging either in terms of time-averaging or in terms of averaging over field configurations.

This equation can be greatly simplified by the averaging procedure and by the following estimation of length scales. On the one hand, the matrix $\tilde{V}_\lambda$ given in (9) evolves in $x$ with a typical correlation length $l_\psi$, which is the correlation length of $\psi$ and which is of the order of the inverse of the width of the momentum distribution. On the other hand, the amplitude of $\psi$ is very small since we consider that the gas has relaxed into a very large box. Thus the elements of $\tilde{V}_\lambda$ are very small, which means that the matrix $\tilde{T}_\lambda(x)$ evolves on a typical length scale $l_T$ which is very large. If the size $L$ of the box in which we have let the gas relax is large enough, the two lengths will obey the Markovian approximation $l_T \gg l_\psi$, which enable to consider a step $dx$ which fulfills

$$l_\psi \ll dx \ll l_T . \tag{11}$$

The second inequality in the above scale hierarchy permits to replace $\tilde{T}_\lambda(x'')$ by $\tilde{T}_\lambda(x)$ in Eq.(10). The first inequality, together with the averaging procedure, has several consequences on Eq. (10). First, one can ignore correlations between $\tilde{T}_\lambda(x)$ and the matrices $\tilde{V}_\lambda(x')$, $\tilde{V}_\lambda(x'')$ since such correlations impact only a negligible part of the integrals. Second, the effect of the first integral averages out since $\langle \tilde{V}_\lambda(x) \rangle = 0$. Finally, in the double integral, one can extend the integral over $x''$ from $-\infty$ to $x'$ since $\langle \tilde{V}_\lambda(t, x') \tilde{V}_\lambda(t, x'') \rangle$ vanishes for distances much larger than $l_\psi$. All the above observations lead to

$$\langle \tilde{T}_\lambda(x + dx) \rangle = \left[ I_d + \int_x^{x+dx} dx' \int_{-\infty}^{x'} dx'' \langle \tilde{V}_\lambda(x') \tilde{V}_\lambda(x'') \rangle \right] \langle \tilde{T}_\lambda(x) \rangle . \tag{12}$$

Using the translation invariance of $\langle \tilde{V}_\lambda(x') \tilde{V}_\lambda(x'') \rangle$ and the fact that we consider an interval $dx \ll l_T$, the above equation reduces to $\partial \langle \tilde{T}_\lambda \rangle / \partial x = \int_{-\infty}^{0} dy \langle \tilde{V}_\lambda(0) \tilde{V}_\lambda(y) \rangle \langle \tilde{T}_\lambda(x) \rangle$. Plugging Eq. 9 into the integrand, this gives

$$\frac{\partial \langle \tilde{T}_\lambda \rangle}{\partial x} = \begin{pmatrix} a_\lambda & 0 \\ 0 & a_\lambda^* \end{pmatrix} \langle \tilde{T}_\lambda(x) \rangle , \tag{13}$$

where $a_\lambda$ reads, in terms of the Fourier components of the field,

$$a_\lambda = \frac{g}{L} \sum_{k,k'} \langle \psi_k^* \psi_{k'} \rangle \int_{-\infty}^{0} dy \, e^{(i(k'-\lambda)+\epsilon)y} , \tag{14}$$

where we have introduced a small positive parameter $\epsilon$, that does not change the result as long as $\epsilon \ll 1/l_\psi$ and that we will let go to zero at the end of the calculation. Invariance under translation of the relaxed system implies that $\langle \psi_k \psi_{k'}^* \rangle = \langle |\psi_k|^2 \rangle \delta_{k,k'}$. We assume moreover that $L$, the box size in which the gas has relaxed, is large enough so that $\langle |\psi_k|^2 \rangle L/(2\pi) = n_\infty(p) = \Pi(p)$. Plugging these results into Eq. (14), replacing the discrete sum by an integral and computing the integral over $y$, we obtain

$$a_\lambda = \frac{g}{L} \int_{-\infty}^{\infty} dk \Pi(k) \frac{1}{i(k-\lambda)+\epsilon} , \tag{15}$$

which leads to

$$a_\lambda = \frac{g}{L} \left( \pi \Pi(\lambda) - i \fint dk \frac{\Pi(k)}{k-\lambda} \right) . \tag{16}$$

Since $a_\lambda$ is independent on position, integration of Eq. (13) simply gives

$$\langle \tilde{T}_\lambda(L) \rangle = \begin{pmatrix} e^{La_\lambda} & 0 \\ 0 & e^{La_\lambda^*} \end{pmatrix} . \tag{17}$$

Coming back to the bare representation by multiplying with $T_{0,\lambda}$ and taking the trace, we obtain the result given in Eq. (5).

# 6 Calculation assuming expansion to the far-field regime

In this section, we derive Eq. 5 using the thought experiment presented in Fig. 1(b): we assume that we let the cloud freely expand during a very long expansion time so that not only the momentum distribution become equal to the rapidity distribution, but the spatial distribution, if expressed as a function of $\frac{x}{t}$ where $x$ is the spatial coordinate and $t$ the expansion time, has become proportional to the rapidity distribution. We will compute the monodromy matrix using the field after the expansion to extract the inverse scattering constants of motion. We assume here that the field density profile is initially centered on $x = 0$ and we use a quantization box which spans the interval $[-L/2, L/2]$ where $L$ is large enough so that the density at the borders of the box is vanishing.

At sufficiently large expansion time, nonlinear effects become negligible since the density is very low. As a result the Fourier components become time-independent, up to the phase factor $e^{ik^2t/2}$. Thus the field is well approximated for long expansion times by

$$\psi(x,t) \underset{t\to\infty}{\simeq} \frac{1}{\sqrt{L}} \sum_k \varphi(k) e^{ikx} e^{-ik^2t/2}, \tag{18}$$

where $\varphi(k)$ does not depend on time. The momentum distribution for such long times is $L|\varphi(k)|^2/(2\pi)$ and is nothing else but the rapidity distribution. Note that we neglect here a phase factor evolving slowly in $\log(t)$ due to the nonlinear term [18]. The quantization box in this section is assumed to be much larger than the size on which the field extends and we replace in the following the sum by an integral. The argument of the exponential terms in the integrand is rapidly evolving in $k$. Making a stationary phase approximation, we obtain, up to a global phase factor,

$$\psi(x,t) \simeq \frac{\sqrt{L}}{\sqrt{2\pi t}} e^{i\frac{x^2}{2t}} \varphi(x/t). \tag{19}$$

In what follows we compute the monodromy matrix using the asymptotic expression of the field given in the above equation.

In order to emphasize the similarity with known quantum physics, let us change representation and introduce the propagator $\bar{T}_\lambda(x) = A_\lambda T_\lambda$ with $A_\lambda = e^{i\frac{x^2}{4t}\sigma_z}$. The evolution equation (6) then becomes $i\partial_x \bar{T}_\lambda(x) = i\bar{V}_\lambda(x)\bar{T}_\lambda(x)$ with

$$i\bar{V}_\lambda(x) = \begin{pmatrix} \frac{1}{2}\left(\lambda - \frac{x}{t}\right) & i\sqrt{\frac{gL}{2\pi t}}\,\varphi^*(x/t) \\ i\sqrt{\frac{gL}{2\pi t}}\,\varphi(x/t) & -\frac{1}{2}\left(\lambda - \frac{x}{t}\right) \end{pmatrix}. \tag{20}$$

Although $i\bar{V}_\lambda$ is not hermitian, this matrix is similar to the time-dependent Hamiltonian of an avoided crossing, the time – not to be confused with the expansion time $t$ which appears in the expression of $\bar{V}_\lambda$ – corresponding to $x$ in the above equation and the crossing occurring for $x = \lambda t$. In this analogy, the diagonal elements of the monodromy matrix correspond to the amplitude associated with diabatic processes. We will indeed use, to compute the diagonal entries of $T_\lambda(L)$, calculations similar to those performed to extract the Landau-Zener formula. More precisely, because of its simplicity, we choose to follow a derivation similar to the one performed in [23].

For the calculation, let us use the same representation as in the previous section, namely let us compute $\tilde{T}_\lambda = T_{0,\lambda}^{-1} T_\lambda$, where $T_{0,\lambda} = e^{-ix\lambda\sigma_z/2}$, such that $\tilde{T}_\lambda(x)$ is stationary in $x$ in regions where the field is vanishing. Since the quantification box is assumed to be very large compared to the extension of the field, on can take the limit $L \to \infty$ for the calculations. Let us denote $c_+$ and $c_-$ the elements of the first column of the propagator $\tilde{T}_\lambda(x)$, whose values at

$x = -\infty$ are $c_+(-\infty) = 1$ and $c_-(-\infty) = 0$. They evolve according to

$$\begin{cases} \frac{\mathrm{d}c_+}{\mathrm{d}x} = \sqrt{\frac{gL}{2\pi t}} e^{i(\lambda x - x^2/(2t))} \varphi^*(x/t) c_-, \\ \frac{\mathrm{d}c_-}{\mathrm{d}x} = \sqrt{\frac{gL}{2\pi t}} e^{-i(\lambda x - x^2/(2t))} \varphi(x/t) c_+. \end{cases} \tag{21}$$

Introducing $u = x/t$, taking the derivative of the first equation and using the second equation, we obtain

$$\ddot{c}_+ = t\left(i(\lambda - u) + \frac{1}{t}\frac{\varphi'^*(u)}{\varphi^*(u)}\right)\dot{c}_+ + t\frac{gL}{2\pi}|\varphi(u)|^2 c_+, \tag{22}$$

where we use the dot notation for derivative with respect to $u$ and $\varphi' = \mathrm{d}\varphi(k)/\mathrm{d}k$. Dividing by $t(\lambda - u)c_+$ and integrating over $u$ we get

$$\int_{-\infty}^{\infty} \frac{\ddot{c}_+}{c_+} \frac{\mathrm{d}u}{t(\lambda - u)} = i\int_{-\infty}^{\infty} \mathrm{d}u \frac{\dot{c}_+}{c_+} + \frac{gL}{2\pi}\int_{-\infty}^{\infty} |\varphi(u)|^2 \frac{\mathrm{d}u}{\lambda - u} + \frac{1}{t}\int_{-\infty}^{\infty} \frac{\dot{c}_+}{c_+} \frac{\varphi'^*(u)}{\varphi^*(u)} \frac{\mathrm{d}u}{\lambda - u}. \tag{23}$$

The last term of the right-hand side is negligible for large enough $t$ since it scales as $1/t$. The first term of the right hand side is computed easily changing the variable $x$ to $c_+$: denoting by $c_+^\infty$ the asymptotic value of $c_+$ at very large $x$ and using the fact that $c_+(-\infty) = 1$, this term gives $i\log(c_+^\infty)$. For the evaluation of the other integrals, let us suppose one approaches the real axis from below in the complex plane, a choice which will be justified afterwards. As in [23], we assume that the function $\ddot{c}_+/c_+$ can be continued analytically in the complex plane and goes to zero at large distances and has no poles, so that the term on the left-hand side vanishes. The second term of the right-hand-side is evaluated using the Sokhotski–Plemelj theorem. Finally, we obtain, using the fact that $L|\varphi(k)|^2/(2\pi) = \Pi(k)$,

$$\log(c_+^\infty) = g\pi\Pi(\lambda) - ig \fint \mathrm{d}q \frac{\Pi(q)}{q - \lambda}. \tag{24}$$

Note that if one would had chosen to estimate the integrals by approaching the real axis from above, then one would have $\log(c_+^\infty) < 0$ so that $|c_+^\infty|^2 < 1$, which is not compatible with the fact that $\det(\tilde{T}_\lambda) = 1$ [18]:[4] together with the fact that the second column of $\tilde{T}_\lambda$ is obtained by permuting the entries of the first column and taking their complex conjugates, the condition $\det(\tilde{T}_\lambda) = 1$ leads to $|c_+(x)|^2 = 1 + |c_-(x)|^2 > 1$.

There are other ways to derive Eq. (24). Following the calculations made in [24] and coming back to the bare representation, one could connect the true solution close to the crossing[5] at $x \simeq \lambda/t$ to the asymptotic solutions at large distance. In such an approach, the principal value integral comes from the effect of the field to second order in $\varphi(k)$ outside the crossing region. Finally, note that the large time expansion was also studied using advance techniques of inverse scattering [18, 26].

Taking the exponential of Eq.(24), we obtain $c_+^\infty$. We come back to the bare representation by mutliplying with $e^{-i\lambda L/2}$, thus obtaining the first diagonal element of the monodromy matrix. Using the fact that the diagonal elements of the monodromy matrix are complex conjugate, and using Eq. (8), we recover Eq. (5).

# 7 Conclusion

The link between the rapidity distribution and the inverse scattering constants of motion, Eq. (5), offers a way to compute the rapidity distribution for a given field configuration $\psi(x)$:

---

[4]Because the columns of $\tilde{T}_\lambda$ are the solutions of the same differential linear equation for two orthogonal initial states, the Wronskian property, together with the fact that $\mathrm{Tr}(\tilde{V}_\lambda) = 0$, imply that $\det(\tilde{T}_\lambda(x)) = \det(\tilde{T}_\lambda(-\infty)) = 1$.

[5]The solution close to the crossing take the from of a parabolic cylindrical function [25].

indeed the inverse scattering constants of motion can be computed once the field configuration $\psi(x)$ at a given time is known. The rapidity distribution, once computed, allows us to predict many interesting features. By definition, it predicts the asymptotic momentum distribution if an expansion is performed. The rapidity distribution shows also its importance when one considers local properties of the system after relaxation: the latter are functional of $\rho(k)$. For instance, one can compute, within the classical field model, local correlation functions after relaxation, adapting results obtained for the Lieb-Liniger model as done in [22]. One can also apply the Generalized Hydrodynamics theory that describes long wave-length dynamics to the classical field model.

Although Eq. (5) has previously been derived using more mathematical approaches, this paper offers an original derivation which does not require knowledge on quantum inverse scattering theory. It might be interesting to explore other methods to derive Eq. (5). One possibility might be to use the fact that the rapidity distribution can be derived from the dynamical structure factor after relaxation [27].

The protocol of section 3 belongs to the class of protocols dubbed quenches, that are protocols where the Hamiltonian is modified suddenly. Many studies investigated the rapidity distribution after a quantum quench in the Lieb-Liniger model [6, 28–30], thus characterizing the system after it has relaxed. The quench considered in section 3 is trivial since the rapidity distribution $\Pi(p)$ is preserved by the quench: the rapidity distribution per unit length $\rho(p)$ after the quench is simply obtained from the initial one by multiplication with $L/L_0$ where $L$ is the length of the system after the quench and $L_0$ its length before the quench.

# 8 Acknowledgments

This work was supported by the ANR Project QUADY - ANR-20-CE30-0017-01. The authors thanks D. Gangardt and J. Dubail for reading the manuscrit.

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
