# Peer review of "Rapidity distribution within the defocusing non-linear Schrödinger equation model"

_SciPost Physics Core, doi:SciPost Phys. Core 6, 064 (2023)_

## Round 1 · Referee Report · Anonymous (Referee 1) · 2023-3-17

Strengths

The paper addresses a very precise and worthwhile problem, namely, that of relating quantities in the Lieb-Liniger model to those in its classical limit, the NLS theory. In particular, the connection they focus on, is the relationship between the rapidity distribution in the quantum model and the monodromy matrix of the classical one.

The fact that the paper is very much focused on one particular question can be seen as a strength, as it is clear from the start what the aims of the work are.

The authors prove their result in two different ways which makes the conclusion more robust.

There is currently a lot of interest in integrable models and their out-of-equilibrium dynamics where the kind of quantities considered here play an important role. There is also increasing awareness of the ability of hydrodynamics to provide a useful description of both quantum and classical theories, so the work is very timely in this respect.

Weaknesses

The main weakness of this paper is that is rather poorly written.
There are just too many grammatical mistakes, misconstructed sentences and words used with some intended meaning which is not the one they normally have in English. I found the paper hard to read mainly for this reason. I think it could be much better written and I will list some possible changes below. However, I would advise the authors to have a read through and see if they can increase clarity of exposition. This will really add to the quality of the paper.

Report

As mentioned earlier, the main result of this paper is establishing a relationship between a quantity which plays a key role in the thermodynamic description of the Lieb-Liniger model (an any other quantum integrable models), namely, the rapidity distribution function, and a quantity (an objects related to it) which plays a crucial role in the description of classical integrable models such as NLS, namely the monodromy matrix that enters the inverse scattering problem. The authors prove this relationship in two different ways.

The calculations that they perform as well as the physical arguments employed, are presented quite clearly.

As I said earlier, I think the work is worthwhile and provides a useful contribution to the field. In particular, better communication between the areas of classical and quantum integrability is worth developing, as their connections are becoming increasingly important in the study of the dynamics and hydrodynamics of integrable systems.

My view is that the paper should be published in SciPost Core as it meets its quality, clarity and originality criteria.

However, the authors should consider improving their writing. I suggest some changes below.

Requested changes

I have quite a few comments, most of them relating to typos, or lack of clarity of exposition although. I am sure I have missed other things so I advise that the authors have a careful reading of the paper. I have also a couple of comments about citations.

1) In the introduction, at the beginning, there is the sentence "The latter are labeled by numbers homogeneous to a momentum, whose number is equal to the particles number, called the rapidities or the Bethe-roots.". I find this sentence rather unclear, in particular I don't know what is meant by "homogeneous to" but I guess it means "proportional to". I think I know what the authors are trying to say. I suggest something like the following: "In finite volume, the latter are labeled by a set of integers which are proportional to the particle's momentum and, in non-relativistic systems, also to the rapidity variable. In non-relativistic integrable models, we call the rapidities Bethe-roots"

2) Later on ".. in the cold atoms field" is better as "... in the field of cold atom experiments"

3) A bit later "in non-linear medium" should be "in a non-linear medium".

4) Throughout the paper the authors use the words "transpose" and "inject" in ways which are not usual in English. When they write in Page 2 "transpose calculations ..." I think what they mean is either "extend" or "generalise" calculations. Similarly "inject" is used several times instead of "substitute" or "plug into", for example "plugging formula (1) into (2)..."

5) In page 2 "on the opposite..." should be "on the contrary..."

6) End of page 2 "for pedagogical purpose" should be "purposes (plural)".

7) The authors use several times the expression "is nothing else but...". It is better to write "is nothing but..."

8) In page 3, they write "...noted \tau_\lambda". Should be "denoted by \tau_\lambda".

9) In page 4 they write "at large \lambda we recover the famous trace identities as it should". Better to write "As expected, for large \lambda the famous trace identities are recovered"

10) Of page 5 they wrote "we thus obtain an expression relying the inverse scattering..." here the word "relying" should be "relating"

11) Later on page 5 "antidiagonal entries" are normally called "off-diagonal entries".

12) On page 6, at the end of the first paragraph it says " after the relaxation in the very large box". Sounds better as "after relaxation in a very large box"

13) A bit later "... a N-points correlation function" should be "an N-point correlation function"

14) Before equation (14) they wrote "a_\lambda writes" should be "a_\lambda reads"

15) Same page a bit later "..by translation of the relaxed..." should be "under translation of the relaxed.."

16) In page 8 I am not sure what the word "homothetic" means. Could the authors make sure this is really what they mean?

17) In page 8 they write "... the quantification box...". Do they mean "the quantization box" in the sense of quantities being quantized (discretized)?

18) In page 9 they write "...denoting c_+^\infty..." should be "... denoting by c_+^\infty..."

19) In the conclusion they write "...permits to predict...". It will be better to say "..allows us to predict.."

20) In the conclusion they write "..the latter are functional of..". Should it be "functions of"?

21) Reference [14] contains many latex characters that have not been correctly processed. This should be corrected.

22) Reference [15] is published in SciPost Phys. 9, 002 (2020)

23) In reference [15] and others some surnames are incorrectly abbreviated. For instance the first author of [15] has surname Del Vecchio and the "Del" cannot be shortened to D. Same goes for "De Luca".

24) Reference [6] is cited for GHD which is correct. However, it is widely known that the theory of GHD was simultaneously developed in two publications and these are normally always cited together. The other one (where they studied spin chains) is

B. Bertini, M. Collura, J. De Nardis, and M. Fagotti, Transport in out-of-equilibrium XXZ chains: exact profiles of charges and currents, Phys. Rev. Lett. 117, 207201 (2016). DOI: 10.1103/PhysRevLett.117.207201

25) In general, I think the authors have not cited lots of literature, especially given the prominent role the Lieb-Liniger model plays in experiments. Perhaps in their introduction they could cite the Newton-Craddle experiment paper, which has become a famous benchmark for the role of integrability in the dynamics of cold atoms and where Lieb-Liniger is found to describe the dynamics of the experiment https://www.nature.com/articles/nature04693

26) Also, since they discuss the connection between Lieb-Liniger and NLS perhaps it can be mentioned that Lieb-Liniger is a very fundamental model of quantum integrability because it has been shown to describe the non-relativistic limit of all known relativistic integrable quantum field theories. This was shown in https://arxiv.org/abs/1608.07548 and https://arxiv.org/abs/1701.06542

  • validity: high
  • significance: good
  • originality: good
  • clarity: good
  • formatting: good
  • grammar: acceptable

Author:  Isabelle Bouchoule  on 2023-05-02  [id 3636]

(in reply to Report 1 on 2023-03-17)

We thank the referee for pointing out that our work is timely and interesting. We thank him/her very much for his/her very careful reading and helping us to improve the writing and making the paper more clear. We really appreciate he or she spend so much time correcting all our mistakes. Below is the answer to his/her comments.

1.What we wanted to say with the formulation "numbers homogeneous to a momentum" was simply the fact that those numbers (real numbers, not integer) have unit mass*velocity. We modified to make this clear.

2.We thank the referee for his/her suggestion and changed the sentence.

3.We corrected the sentence. Thank you for the correction.

4.We change changed the sentences with the word "transpose" or "transposable". We also remove the occurences of "inject" or "injecting"

5.We made the change.

6.We applied the correction.

7.We now write "is nothing else but"

8.We made the change.

9.We thank the referee for his/her reformulation.

10.We agree the use of "relying" was improper in this context.

11.We changed to "off-diagonal"

12.We removed the "the" before the word "relaxation". However, we preferred to keep the "the" before the word "box". The reason is that it refers to the bow we introduced in the sentence above.

13.We made the modification.

14.We made the change.

15.Modification done.

16.We removed the word "homothetic" and say "[..] the spatial distribution, if expressed as a function of x/t where x is the spatial coordinate and t the expansion time, has become proportional to the rapidity distribution."

17.Done.

18.Done.

19.Done.

20.The local properties are function of the whole function $k\rightarrow\rho(k)$. We think the appropriate term is then "functional". We kept this terminology.

21.Done.

22.We modified the reference. We are sorry for this mistake.

23.Thank you for correcting. We modified the reference.

24.We agree and we are sorry we did not properly give the references to GHD before. We included the new reference.

25.We thank the referee to suggest to emphasize the relevance of the Lieb-Liniger model to describe experiments. We modified the begining of the introduction to emphasise this point. We cite the recent review paper, whose I. Bouchoule is one of the authors, which reviews the important experimental tests of the Lieb-Liniger model in cold atoms experiment. Of course, we are aware of the Nature paper entitled "a quantum Newton's craddle" and we agree it played a huge role in the subsequent theoretical development on the understanding of non-equilibrium physics in integrable systems. We are not convinced it is very much related to this particular paper, but we have no problem to cite it, and we did. The sentence added in the introduction to answer referee's comment is : "On the experimental point of view, it describes remarkably well cold-atoms experiments~(see for instance the review [J. Stat. Mech. 2022(1), 014003 (2022)]), among them the famous Newton's Craddle experiment [Nature 440(7086), 900 (2006)]."

26.We thank the referee for giving us these interesting references that emphasizes the key role of the Lieb-Liniger model. We cite them in the beginning of the introduction, with the sentence "On the theoretical point of view, it is a paradigmatic integrable model, that is the non-relativistic limit
of all known integrable quantum field theories [J. Stat. Mech. 2016(12), 123104 (2016), J. Phys. A: Math. Theor. 50(23), 234002 (2017)]."

Author:  Isabelle Bouchoule  on 2023-05-02  [id 3635]

(in reply to Report 1 on 2023-03-17)

We thank the referee for pointing out that our work is timely and interesting. We thank him/her very much for his/her very careful reading and helping us to improve the writing and making the paper more clear. We really appreciate he or she spend so much time correcting all our mistakes. Below is the answer to his/her comments.

1.What we wanted to say with the formulation "numbers homogeneous to a momentum" was simply the fact that those numbers (real numbers, not integer) have unit mass*velocity. We modified to make this clear.

2.We thank the referee for his/her suggestion and changed the sentence.

3.We corrected the sentence. Thank you for the correction.

4.We change changed the sentences with the word "transpose" or "transposable". We also remove the occurences of "inject" or "injecting"

5.We made the change.

6.We applied the correction.

7.We now write "is nothing else but"

8.We made the change.

9.We thank the referee for his/her reformulation.

10.We agree the use of "relying" was improper in this context.

11.We changed to "off-diagonal"

12.We removed the "the" before the word "relaxation". However, we preferred to keep the "the" before the word "box". The reason is that it refers to the bow we introduced in the sentence above.

13.We made the modification.

14.We made the change.

15.Modification done.

16.We removed the word "homothetic" and say "[..] the spatial distribution, if expressed as a function of x/t where x is the spatial coordinate and t the expansion time, has become proportional to the rapidity distribution."

17.Done.

18.Done.

19.Done.

20.The local properties are function of the whole function $k\rightarrow\rho(k)$. We think the appropriate term is then "functional". We kept this terminology.

21.Done.

22.We modified the reference. We are sorry for this mistake.

23.Thank you for correcting. We modified the reference.

24.We agree and we are sorry we did not properly give the references to GHD before. We included the new reference.

25.We thank the referee to suggest to emphasize the relevance of the Lieb-Liniger model to describe experiments. We modified the begining of the introduction to emphasise this point. We cite the recent review paper, whose I. Bouchoule is one of the authors, which reviews the important experimental tests of the Lieb-Liniger model in cold atoms experiment. Of course, we are aware of the Nature paper entitled "a quantum Newton's craddle" and we agree it played a huge role in the subsequent theoretical development on the understanding of non-equilibrium physics in integrable systems. We are not convinced it is very much related to this particular paper, but we have no problem to cite it, and we did. The sentence added in the introduction to answer referee's comment is : "On the experimental point of view, it describes remarkably well cold-atoms experiments~(see for instance the review [J. Stat. Mech. 2022(1), 014003 (2022)] ), among them the famous Newton's Craddle experiment~[Nature 440(7086), 900 (2006]."

26.We thank the referee for giving us these interesting references that emphasizes the key role of the Lieb-Liniger model. We cite them in the beginning of the introduction, with the sentence "On the theoretical point of view, it is a paradigmatic integrable model, that is the non-relativistic limit
of all known integrable quantum field theories~\cite{bastianello_non_2016,bastianello_non_2017}."

---

## Round 1 · Referee Report · Anonymous (Referee 2) · 2023-4-3

Strengths

  • interesting new derivation of a known result

Weaknesses

  • previous results not correctly acknowledged
  • presentation of the derivations quite unclear

Report

The authors want to re-derive a known result connecting integrals of motion in NLS theory to rapidity distribution by means of a very nice idea involving the expansion of a gas in a free trap. The derivation is nice and calculations seem all correct, however there are major issues in referencing to previous works, in correctly acknowledging that the result they derive is already known and so on. In particular

  • Regarding the main result: eq 5 has been written first in https://iopscience.iop.org/article/10.1088/1742-5468/2016/06/064011 (eq 424) and then also re-used in https://www.scipost.org/SciPostPhys.9.1.002/pdf (eq 76-78). The author must be very clearly indicate that the result is known (so this claim “Although the results derived in this paper are probably known by specialists of classical integrable systems, to our knowledge, the formula we give in this paper, Eq. (5), is not found as it stands in previous literature” is false) and that in their work they re-derive a known equation by other means. Failure to explicitly rewrite this part and to not include careful references can be judged as scientific misbehaviour.

  • reference to GHD theory is missing Bertini et al 2016 https://journals.aps.org/prl/abstract/10.1103/PhysRevLett.117.207201

  • quenches in Lieb Liniger have been considered in many different papers, I invite the authors to cite some of them, for example https://arxiv.org/abs/1505.03080 https://arxiv.org/abs/1308.4310 https://arxiv.org/abs/1509.08234?context=cond-mat https://journals.aps.org/pra/abstract/10.1103/PhysRevA.91.051602 [this latter very important, relating conserved quantities to distribution of rapidities]

  • equation 8: what it L? where is time dependence? in psi(x,t) ? Quantities should be defined with much more clarity

Requested changes

see report

  • validity: good
  • significance: good
  • originality: good
  • clarity: good
  • formatting: acceptable
  • grammar: reasonable

Author:  Isabelle Bouchoule  on 2023-05-02  [id 3634]

(in reply to Report 2 on 2023-04-03)

We thank the referee for his careful reading of our manuscript. We appreciate that he/she finds our work interesting as being a new way of presenting known results.
We thank him for pointing out that the results were known. We are sorry we did not sufficiently acknowledge previous works.

-We were aware of the two papers he/she refers to: A. D. Luca and G. Mussardo, "Equilibration properties of classical integrable field theories", J. Stat. Mech. 2016(6), 064011 (2016) and G. del Vecchio del Vecchio, A. Bastianello, A. De Luca and G. Mussardo, "Exact out-of-equilibrium steady states in the semiclassical limit of the interacting Bose gas", SciPost Physics 9(1), 002 (2020)". Both papers were cited in our submitted paper (see the 3rd paragraph of the introduction).
We now refer to them again after our Eq. (5) and give more details. Concerning ref [J. Stat. Mech. 2016(6), 064011 (2016)], we now give the two equations of [J. Stat. Mech. 2016(6), 064011 (2016)] which are the equations corresponding to our Eq. (5) for the Sh-Gordon model. It is our lack of knowledge in the field of integrable systems that prevent us to immediately do the link with our result, that holds for the Lieb-Liniger model.
Concerning the ref [SciPost Physics 9(1), 002 (2020)], we were aware of its eq. (76) corresponding to our eq.(5). However, we find that both equations agree only in the limit of large $\lambda$. In this limit, the $p$ that contribute to the principal value integral in our eq.(5) are small compared to $\lambda$ and one can expand $1/(p-\lambda)$ in series in p. Then, Eq. (76) of [SciPost Physics 9(1), 002 (2020)] is recovered. But we do not think this is true for smaller $\lambda$. What is the opinion of the referee ?
In the present version of our paper we put the sentence : "Eq. (5) also coincides with the formula (76) of [del Vecchio del Vecchio, SciPost Physics 9(1), 002 (2020)] at large $\lambda$"

-We agree that one should cite both the reference [Castro-Alvaredo et al., Phys. Rev. X 6(4), 041065 (2016)] and the reference [B. Bertini et al. , Phys. Rev. Lett. 117(20), 207201 (2016)] when referring to GHD theory. We are sorry for forgetting a reference, we added it.

-We added references concerning quenches in the Lieb-Liniger model in the last paragraph of the conclusion.

-We agree that the reference [F. H. L. Essler et al. , Phys. Rev. A 91(5), 051602 (2015)] is very important to understand the crucial role of the rapidity distribution. We added the reference in the introduction.

-We modified the section 4 to improved its clarity.
*We added the following sentence at the beginning of the section : "We consider a field $\psi(x, t)$ whose time evolution is given by the NLSE Eq. (2) and which obeys periodic boundary conditions on a box of length L."
*We emphasise that $\psi$ depends on time and we explain our choice to omit the time in the equations for the derivation of $\tau_\lambda$ using the following sentence : "At any time t, one can compute these constants of
motion knowing the field configuration at the time t. Thus in the following we consider the
one-dimensional function $x \rightarrow \psi(x, t)$ and we omit the time variable."
*Note that, between Eq. (6) and Eq.(7), there is the sentence "the monodromy matrix depends on time via the time dependence of $\psi(x)$." We think that, together with the modifications done to the begining of this section (see the two points above), the time dependency is now clear.

---

## Round 1 · Referee Report · Anonymous (Referee 3) · 2023-4-6

Strengths

  • physically motivated derivation of a relation between the asymptotic momentum distribution and generating function of conserved charges from the inverse scattering method

Weaknesses

  • not enough precision in the presentation of the results

Report

The work has a scientific merit and certainly is worth communicating after fixing the points raised in the reports. I agree with the reports of the other two referees and additionally I would like the authors to consider some point listed below. I suggest a major revision to address all these issues.

Requested changes

  • please rephrase the last sentence of the abstract to improve its clarity.

  • whereas the work is focused on the NLS I think it's certainly worth citing the following papers, S. Negro and F. Smirnov, Nucl. Phys. B 875, 166 (2013); S. Negro, Int. J. Mod. Phys. A 29, 1450111 (2014) which constitute a fundamental work making the results of [5] possible.

  • the rapidity distribution in LL gas can be also determined from correlation functions as shown in J. De Nardis et al SciPost Phys. 3, 023 (2017)

  • before eq. (4) the authors write "This conserved distribution is nothing else but the rapidity distribution". At this point is not very clear what authors mean by the rapidity distribution, is it from the full quantum LL model? Some explanation and reference to relevant works would potentially help the reader understand this.

  • validity: high
  • significance: good
  • originality: high
  • clarity: ok
  • formatting: good
  • grammar: good

Author:  Isabelle Bouchoule  on 2023-05-02  [id 3633]

(in reply to Report 3 on 2023-04-06)
Category:
validation or rederivation

We appreciate that the referee thinks that our work "has a scientific merit and is worth communicating". We thank him for his remarks. Below is an answer to his comments.

-We rephrased the last sentence of the abstract, to make it more clear.

-We thank the referee for the references about the derivation of the one-point correlations functions in the Sh-Gordon model. We included these references in a footnote.

-We thank the referee to point the interesting fact that the rapidity distribution could also be defined via the dynamical structure factor. We added a sentence at the end of the second paragraph of the conclusion, with the reference to the paper of J. de Nardis, to point this and suggest that this could lead to another way of deriving the rapidity distribution.

-Rapidities are usually defined from the Bethe Ansatz and their classical counterpart are obtained either via quantum inverse scattering and Algebraic Bethe-Ansatz or by identifying the classical counterpart of the Bethe eqations equations. In this paper however we use the definition of the rapidity distribution as being the asymptotic momenta after expansion to very large distances. This is explained in the introduction, and particularly in its forth paragraph. Before Eq. (4) we now refer to the introduction. We also add after Eq. (4) the following sentence: " This equality provides a definition of the rapidity distribution, which is that used in this paper." We hope the status of Eq. (4) is now more clear.

---

## Round 2 · Referee Report · Anonymous (Referee 1) · 2023-5-23

Strengths

Same as on my first report

Weaknesses

Same as in first report except that the authors have made improvements following my suggestions. Most of my criticism was about the paper not being very clear in terms of writing style. This has now been improved.

Report

The authors have engaged with my comments and made improvements accordingly. Since my comments were not critical of the main results but rather or their presentation and the presentation has now improved, I am now happy with the submission and recommend publication in its current form.

---

## Round 2 · Referee Report · Anonymous (Referee 2) · 2023-6-2

Strengths

Same as previous report

Report

the authors fulfilled all my requests, the paper can now be published

---

## Round 2 · Referee Report · Anonymous (Referee 3) · 2023-6-15

Report

I'm satisfied with the improvements made by the authors and recommend the manuscript for the publication.

---

## Round 2 · Author Response

Dear editor, dear referees,

We thank the referees for their work on our manuscript and their intersting remarks. Please see the answer we provide on the Scipost webpage to each referee report. Each point raised by the referee has been considered and the paper has been modified accordingly.

We hope the referees will agree with our replies,

Best regards,
The authors.

---

## Round 2 · List of Changes

• Last sentence of the abstract has been modified to be more clear -The beginning of the introduction has been modified and expanded. We emphasize the importance of the Lieb-Liniger model both for its experimental relevance and its theoretical key role. We added corresponding references. -In the introduction, we replaced "homogeneous to a momentum" by "whose unit is mass x velocity " -Before Eq. 4, we added "As explained in the introduction ...". After the equation, we added a sentence to emphasize we use EQ. 4 as the definition of the rapidity distribution. -After Eq. (5), we extended the discussion to compare to previously published results. In particular, we discuss the link with the results of the references [A. D. Luca and G. Mussardo, J. Stat. Mech. 064011 (2026)] and [G. del Vecchio del Vechio et al., Scipost Physics 9, 002 (2022)]. -At the beginning of section 4, we recall that we consider a field with periodic boundary conditions on a box of length L. -Before Eq. (6), we added the sentence "Thus in the following we consider the one-dimensional function $x\rightarrow \psi(x,t)$ and we omit the time variable." -We modified the second paragraph of the introduction. We now write that "Eq. (5) has previously been derived using more mathematical approaches". We also mention at the end of this paragraph that one could explore the possibility to use the link between the rapidity distribution and the dynamical structure factor to provide an alternative calculation of Eq. (5). -We added a new paragraph to the conclusion that makes the link between the thought experiments investigated in this paper and quenches protocols previously considered within the Lieb-Liniger model. -We made several corrections of English, following the advises of the referees.

---

## Editorial Decision

published